# Clear Aligners: Between Evolution and Efficiency—A Scoping Review

**DOI:** 10.3390/ijerph18062870

**Published:** 2021-03-11

**Authors:** Alessandra Putrino, Ersilia Barbato, Gabriella Galluccio

**Affiliations:** Department of Oral and Maxillofacial Sciences, “Sapienza” University of Rome, 00161 Rome, Italy; ersilia.barbato@uniroma1.it (E.B.); gabriella.galluccio@uniroma1.it (G.G.)

**Keywords:** clear aligners, orthodontics, biomechanics, efficiency, efficacy, attachments, divots, auxiliaries, mini-screws, elastics

## Abstract

In recent years, clear aligners have diversified and evolved in their primary characteristics (material, gingival margin design, attachments, divots, auxiliaries), increasing their indications and efficiency. We overviewed the brands of aligners used in Italy and reviewed the literature on the evolution of clear aligners based on their characteristics mentioned above by consulting the main scientific databases (PubMed, Scopus, Lilacs, Google Scholar, Cochrane Library). Inclusion and exclusion criteria were established. The data were collected on a purpose-made data collection form and analyzed descriptively. From the initial 580 records, 527 were excluded because they were not related to the subject of the review or because they did not meet the eligibility criteria. The remaining 31 studies were deemed comprehensive for the purpose of the review, although the “gingival margin design” feature and “auxiliaries” tool are not well represented in the more recent literature. Current knowledge on invisible aligners allows us to have a much clearer idea of the basic characteristics of aligner systems. There remains a need to deepen the use of systems other than Invisalign™ to give greater evidence to aligners that are very different based on the characteristics analyzed here and that are very widespread on the market.

## 1. Introduction

Clear aligners were introduced in the United States, where they were born at the end of the 1990s by the company Align Technology© (Santa Clara, CA, USA), which then gave life to the Invisalign^®^ system, and then they were distributed in Italy and other European countries starting from 2001 [1,2]. Since then, the interest and the diffusion of this therapeutic alternative to the classic multibracket orthodontic therapy have increased exponentially [3,4,5], also considering the needs and aesthetic perception of orthodontic patients [6]. Almost 20 years later, the commercial proposals regarding aligners include many different brands all over the world (almost twenty just in Italy), and their indications, applications and constitutive features have evolved [7,8,9]. The thermoplastic materials used, the gingival margin design, and the different strategies used to guide orthodontic movement such as attachments, divots and auxiliary tools converge in determining the effectiveness of a system of aligners [10]. Clear aligners do not seem to all be the same and consistent differences can be observed between different brands, as if we can no longer speak of a “single system” but of “several different systems” [11,12]. Their indication was initially limited to the leveling and alignment of the arches in the presence of slight crowding or diastemas [13,14]. Today, a large part of cases, even moderately or extremely complex, can be managed with aligners. The entire resolution of a case in which aesthetic needs can also be associated with functional ones can be often successfully treated with the so-called “hybrid” therapies, where clear aligners’ action is joined to auxiliary tools (mini-screws, elastics, sectional wires, rapid palatal expanders, etc.) [15,16,17,18,19]. The quality of the orthodontic digital workflow, also extended to other specialized fields [20,21,22], has made it possible to better evaluate the fit of the aligners and their overall design for better case management [23,24,25]. The purpose of this scoping review is to systematically map the scientific evidence and any gaps in knowledge on the evolution of invisible aligners available in Italy on the basis of their primary characteristics (thermoplastic material, gingival margin design, the presence of attachments with different shapes, the alternative use of divots and auxiliary tools such as elastics and mini-screws, etc.), which are highlighted by the manufacturers as winning points of their clear aligners’ effectiveness compared to competitors.

## 2. Materials and Methods

Preliminary research on clear aligner systems available in Italy has been performed. Technical information and data were searched on official websites of the manufacturers and in professional magazines. The primary characteristics have been identified: materials used, design of the gingival margin, presence of attachments or other strategies to guide orthodontic dental movement. The research of the articles useful for the scoping review was last conducted on 5 January 2021 to evaluate the evidence in the scientific literature given to the aligner systems available on the Italian market as an alternative to Invisalign. Two independent operators (A.P. and G.G.) consulted the English language literature available for full-text reading on the scientific literature databases: PubMed, Scopus, Lilacs, Cochrane Library, following the framework for scoping reviews of the PRISMA-ScR guidelines [26]. The research questions were: “what is the scientific evidence of the clear aligner systems available in Italy as alternative to Invisalign?” and “are their constitutive characteristics comprehensively described?”. The potentially relevant articles were selected following these eligibility criteria: published in the period of 2015–2021, written in the English language, abstract and full-text available, investigating the primary characteristics of clear aligners. No limit was put on study design. Studies on Invisalign invisible aligners were included only if used in comparative studies with other brands of aligners present in the results of our preliminary market research, or if significantly related to the characteristics that are the subject of this review. Two consecutive searches have been performed using “AND” and “OR” Boolean operators between free text terms or keywords combined as follows:Clear aligner appliancesOrthodontic appliance, removable1 OR 2 ORTooth movementBiomechanicsAttachmentsDivotsMini-screwsElasticsAuxiliariesOR 4 OR 5 OR 6 OR 7 OR 8 OR 9 OR 103 AND 11

After comparing and unifying the results found by the two operators, duplicate results of the different databases were deleted and a third reviewer (E.B.) read the abstracts of all the results in order to verify that the articles properly adhered to the objectives of the study. Articles that have not been deemed adequate since this first observation have been discarded. Doubtful articles have been retained to reserve an assessment for a full-text reading. The three reviewers, always independently, extracted data in duplicate and subsequently compared them. The following information was extracted: authors, year, country, study design (reviews were included), sample size and characteristics (subjects, dental models, clear aligners, brands used), main topic of the article, type of primary feature of clear aligner investigated, important conclusions/outcomes. Subsequently, all the authors analyzed these results in agreement in order to be able to discuss them in this article. Additionally, the scoping review search strategy was depicted as a flow diagram using the PRISMA model [27] for systematic reviews.

## 3. Results

The preliminary research allowed us to know the basic distinctive features of the clear aligners which are used more in Italy (name, manufacturer, material, introduction year, wearing time, indications, contraindications, the presence of attachments and/or divots, auxiliaries, number of studies). Of 19 brands for which it has been possible to describe the constitutive characteristics, only eight, in addition to Invisalign, are mentioned in the official scientific literature since they were used in experimental studies (Table 1). Invisalign has the highest rate of presence in the literature with 110 papers, followed by Clear Aligners with 10 papers, F22 (Sweden & Martina SpA, Due Carrare, Italy) with 9 research articles, Airnivol (AirNivol SpA, Navacchio, Italy) with 3 articles and Nuvola (Nuvola®, Vicenza, Italy) with 2 articles, and ALL IN, Arc Angel, Smiletech and Sorridi with 1 research article only. For the scoping review, instead, we identified 580 studies of potential relevance. After the removal of duplicated results from the different databases (*n* = 443), 137 articles were screened in detail, and 52 of these were considered eligible for a full-text review. Of these, 31 studies published between 2015 and 2021 were included in the scoping review. The flow diagram (Figure 1) describes the entire search strategy and review process. Fifteen articles are dedicated to the study of materials and their properties, twelve to attachments and other movement strategies, two refer to the design of the gingival margin which, however, is not really the main object of the articles in question (in fact, they deal specifically with the other two already mentioned topics taken into consideration in this study), and five studies help determine the usefulness of auxiliary tools in combination therapies with aligners (Table 2). There are sixteen in vitro comparative studies, while there are two digital models. There are four prospective clinical studies, two retrospective studies, three case reports and one case series. There are three reviews considered useful for the compilation of the scoping review; one of them is a systematic review with meta-analysis, the other two are narrative reviews.

The most active country in research on the evolution of aligners based on the primary characteristics object of this review of the literature is Italy, which, with twelve studies, is also the one to have carried out most of the clinical research (eight out of ten among prospective works, retrospectives, case reports and case series).

## 4. Discussion

Since the introduction of the aligners with the Invisalign™ brand distributed by the US company Align Technology© [1,2], the commercial offer of the aligners has been significantly enriched with national, European and international competitor brands. They have also diversified their characteristics over time, defining a current trend that differs from Invisalign™ for the modification of some essential characteristics, ranging from the type of material used to the design of the gingival margin that is increasingly straight and extended beyond the gingival zenith to give greater adherence and reduce the presence of attachments for retentive purposes. The presence of attachments is questioned by some brands of aligners who propose their almost total absence and the introduction of other tools for controlling movements, such as divots, which are currently only actively used by three manufacturers. The use of auxiliaries such as elastics or mini-screws has certainly broadened the indications for the use of aligners for orthodontic therapies, even of a certain complexity that the aligners alone would not be able to manage in a predictable way. Thermoplastic material, of which the aligners are made, their design at the gingival level and the possibility of using attachments or alternative movement strategies such as divots and auxiliaries are precisely the primary characteristics on which the effectiveness and efficiency of the various aligner systems depend. The concepts of “efficacy and efficiency” in clinical orthodontics are used interchangeably to describe “achieving the desired results without wasting time for the orthodontist and the patient” [10]. In the field of invisible aligners, the search for ever better efficacy and efficiency is expressed through the ability of these devices to perform more or less complex dental movements in a predictable way as much as traditional fixed appliances with equally stable results [3]. These elements are linked to the ability to maintain adequate adhesion of the aligner (fitting) on the teeth, but also to guarantee the transmission of the forces necessary to move the teeth in a predictable way without sacrificing the comfort of the patient, which helps with compliance [4,5]. Despite the wide commercial offer, the scientific literature lacks studies that testify to the use of alternative brands with different systematics. There are few comparative studies between systematics, and more than 90% of publications focus on the use of Invisalign™ exclusively. In any case, based on the data observed in this scoping review, it seems that Italy is the country that joins, more than others, the scientific interest in alternative brands to Invisalign with the study of the primary characteristics of each of them. Eight brands are mentioned in the scientific literature. Airnivol (AirNivol S.p.A, Navacchio di Cascina -Pisa, Italy) has been used in three comparative studies between systematics on fitting influenced by the thickness of the aligner, by exposure to the oral environment and in the study of the extrusion movement of the central incisor [29,30,53]. ALL IN (Micerium S.p.A., Avegno- Genova, Italy) and Arc Angel (Gruppo Dextra, Modena, Italy) have only been used in one study about the comparative analysis of the gap and thickness of different brands [27]. The aligner named Clear Aligner (Scheu Dental GmbH, Iserlohn, Germany) has been used in ten studies on the material (PET-G) they are made of and on its mechanical properties after the thermoforming process, exposure to saliva and different beverages, after stress relaxation due to use and cito-toxicity [29,30,31,32,33,34,54,55,56,57]. F22 is a system of clear aligners (Sweden & Martina SpA, Due Carrare—Padova, Italy) used for experimental research in nine studies. Two of them explore the aligner fitting based on the presence of attachments compared with clear aligners produced by different companies [27,55]. One comparative study evaluates the stress relaxation after use [32]. Other studies evaluate the behavior of the clear aligner after oral exposure and with aging [58,59]. A clinical study has been published on the efficiency of the use of this clear aligners’ system with elastics in class II malocclusions [15]. A recent study evaluates the progress of the therapies with this clear aligner and other brands during the last pandemic [35]. Another two studies show how to plan hybrid therapies with clear aligners and fixed appliances to manage class II and III malocclusions [36,37]. Nuvola (G.E.O. S.r.l., Rome, Italy) is present in the scientific literature with two published studies. It has been used in the comparative study by MicroCT X-ray on the aligners’ gap and thicknesses [27], and in another recent study on the predictability of this aligner during the movement of anterior teeth [38]. The use of the clear aligners named Smiletech (Ortodontica Italia s.r.l.) and Sorridi (Tecnologia Dentale, Latina, Italy) is documented in the study describing the advantage of using clear aligners when regular orthodontic checks cannot be performed in person. The Sorridi system includes divots (and no attachments), and this is probably the reason why patients wearing these clear aligners reached better results and had no discomfort during the SARS-Cov2 pandemic [35]. Clear aligners are made of thermoplastic materials. The most commonly used materials are polyurethane, polyester and polyethylene glycol terephthalate (PETG). Many spectrophotometric studies analyze their composition to confirm the chemical structure stated by the manufacturers and eventual differences in different brands using the same material [33,39]. This kind of article compares the clear aligners’ material mechanical properties with experimental studies in vitro, mainly applying the test indentation (with Vickers indenter), the Martens hardness, the indentation modulus, the elastic to total work ratio (elastic index) and the indentation creep [40]. The clear aligners made in polyurethane showed higher hardness and modulus values, a slightly higher brittleness and lesser creep resistance compared with the PETG-based products. The results offered are very important because they anticipate the clinical behavior expected in clear aligners of different brands, motivating qualities and limitations [27,33,39,40]. Many studies evaluate the stability of the materials after their average use of two weeks through the colorimetric alterations of the aligner. Studies of this type simulate the environment of the oral cavity through solutions of artificial saliva and exposure to highly pigmented foods such as wine, coffee, cola, tea and different cleaning methods [41]. The most recent data, in which both PET-G and polyurethane based aligners are used, agree in affirming that there are foods that stain more than others (above all black tea), that polyurethanes are more subject to color alterations, that even the surface analysis with a scanning electron microscope shows a greater alteration of the integrity of the material, and that this becomes all the more significant the longer the aligner is used (the aligners after two weeks show values of surface alteration above the threshold level of 0.20 μm) [30,42]. Other authors [31,32,43] found that material properties change during the wear-time, and this may affect treatment outcomes since the intraoral aging of clear aligners (regardless of the type of material they are made of) through biofilm modifications and oral environmental conditions might have an adverse effect on material properties and stability over the treatment time, compromising the force delivery capacity and treatment efficacy. A final aspect to analyze concerning the materials and their properties is the thickness of the thermoplastic material with which the invisible aligners are manufactured. The thickness of the aligners is predetermined by some manufacturers or is variable according to others, and in some system it is established based on the type of treatment or used alternately to apply forces of variable intensity that simulate fixed orthodontics [28,29,34,44,45]. The production process seems to affect the final thickness of the aligners with effects that can be negative if the creation takes place through 3D printing [44], even if the process of production is time-saving and ensures an aligner which is mechanically stronger and more elastic than the conventionally produced thermoplastic-based thermoformed clear dental aligners [29]. The intraoral use may change the thickness but not in a clinically relevant manner, and the thermoforming process does not induce an alteration of the active or passive aligner configurations [45]. Furthermore, the thickness (commonly used 0.5, 0.625, 075 mm) does not affect forces and moments generated under the most commonly examined tooth movements (rotation and tipping) [28], while labial or palatal movements are negatively affected by the incremented thickness of the clear aligners (recommended by some manufacturers during the setup) [34]. The ability of an invisible aligner to move teeth is given by the pressure exerted by the material on the tooth. Traditionally, the displacement is guided by the presence of composite resin buttons which are applied to the buccal and/or oral-palatal surface of one or more teeth, and whose shape and position depend on the function they must perform [14,46]. Attachments have also undergone some sort of evolution over the years. Initially, they were only ellipsoidal in shape: horizontal for active intrusion movements and vertical for retentive purposes. We then moved on to rectangular ones distinguished by horizontal, vertical and beveled shapes [47]. The horizontal ones are for intrusions and extrusions on premolars and incisors, on premolars to increase aligner stability when using class II or III elastics, for retention in subjects with short dental crowns or hypodivergent patterns (to control the Spee curve and deep bite), and on restored teeth because the attachment’s bonding area is smaller. The vertical rectangular attachments are used to derotate the canines and premolars (in particular, on the mandibular ones to close the extraction spaces), for the axial control of the anterior teeth and for the uprighting of the posterior teeth (for example, after extractions, in pre-prosthetic orthodontics or in the preparation of implant sites). The beveled rectangular attachments are used in cases of deep bite; in particular, they are indicated in the second classes second divisions and to extrude canines and incisors [12,16]. The beveled attachments have then evolved in turn into customized and smaller beveled attachments that have the same function as conventional ones but are applied along the vertical axis of the tooth in the so-called “active surface area” to avoid the sliding soap effect during placement. Always customized according to the shape, length and width of the teeth are the teardrop attachments that are used in cases of multiple rotations (greater than five elements), to guide the derotation of the canines and when, in general, the correction exceeds 2 degrees [48]. More recently introduced are “power ridges”, initially introduced only for the “teen” treatments of Invisalign™ and now extended to all treatments in which it is necessary to improve the correction of the torque (>3°) and the vertical control of the axis of the incisors, but also in multiple movements, for example, in a second class second division before extruding [17,38]. Finally, the “bite ramps” are horizontal attachments that are applied to the lingual surface of the upper teeth to correct deep bites and are applied buccally only in cases of cross bite [16,49]. In all cases, the effectiveness of the aligner–attachment–tooth interaction depends a lot on the precision with which the operative protocol used for bonding the attachment itself is performed and on the composite material used [46,50]. The divots are small depressions programmed and pre-inserted on the invisible aligners that are able to replace the attachments to guide many movements (rotations, minor tipping movements, buccal-oral movements) and/or guarantee the retention of the aligner. In the literature, their action is still poorly documented [12,27], despite their very promising efficacy even in cases whose management can be very complex, such as the recovery of a compromised element due to excessive torque with bone defect [51]. Although the design of the gingival margin of the aligner is important in defining the characteristics of one aligner rather than another, this aspect in the literature has so far been treated only marginally [12,27]. The studies that mention this element refer to a single study that has actually developed research on this topic and which is present in the literature, but dates back to 2012 [60]. Based on it, the aligners can also be classified according to the design of the gingival margin, which can be scalloped by reproducing the patient’s normal gingival scallop, straight at the gingival zenith level, or straight but extended 2 mm beyond the gingival zenith. These three types of margin would affect the retention of the aligner (the most retentive aligner margin design is the straight-line margin cut 2 mm above the gingival zenith), but also the fitting of the aligner on the tooth and, because of that, the predictability of the clear aligner therapies [27]. The versatility of an aligner is also given by the possibility of extending its indications by inserting auxiliary elements in the system, such as mainly elastic and mini-screws [12]. Not all systems on the market include the use of these integrated devices. Intermaxillary elastics are used, with good movement control and allowing the maintenance of adequate oral hygiene, mainly for class II corrections and dental cross bites by making cuts on the aligners that allow the bonding of orthodontic buttons or by making changes on the aligner itself, which can act as a direct anchor for the elastic bands [15]. A recent study states “no significant Class II correction or overjet reduction was observed with elastics” and “additional refinements may be necessary to address problems created during treatment, as evidenced by a posterior open bite incidence” [52]. Orthodontic mini-screws can be used in hybrid therapies to aid both skeletal maxillary expansion in class III malocclusions [36] and the treatment of class II malocclusions to achieve unilateral distalization by means of a single bone-borne appliance followed by the treatment with invisible aligners [37].

## 5. Conclusions

The most recent contributions to the scientific literature on the basic constitutive characteristics of invisible aligners show that the knowledge of experts converges in classifying the behavior of materials and their mechanical properties, allowing the establishment of advantages and disadvantages of the different brands of aligners. Even on movement strategies, the variety of attachment types allows us to build an increasingly precise clinical setup. The possibility of using divots to support attachments or in the total replacement of them is still poorly documented. Even the design of the gingival margin is not an element whose influence on the effectiveness of the aligners is now well documented and clarified. The use of auxiliary tools is documented in studies with few observations. These results and the few comparative studies between systems of invisible aligners place the attention on the need to deepen through experimental studies those systems which, although very popular and widespread in clinical practice, are not supported by scientific data. The need for this is also given by the fact that many of them in their basic constitutive characteristics are not superimposable to the systems on which most of the scientific experimentation documented by the literature is concentrated, and this leads to an important gap between knowledge and clinical practice.

## Figures and Tables

**Figure 1 ijerph-18-02870-f001:**
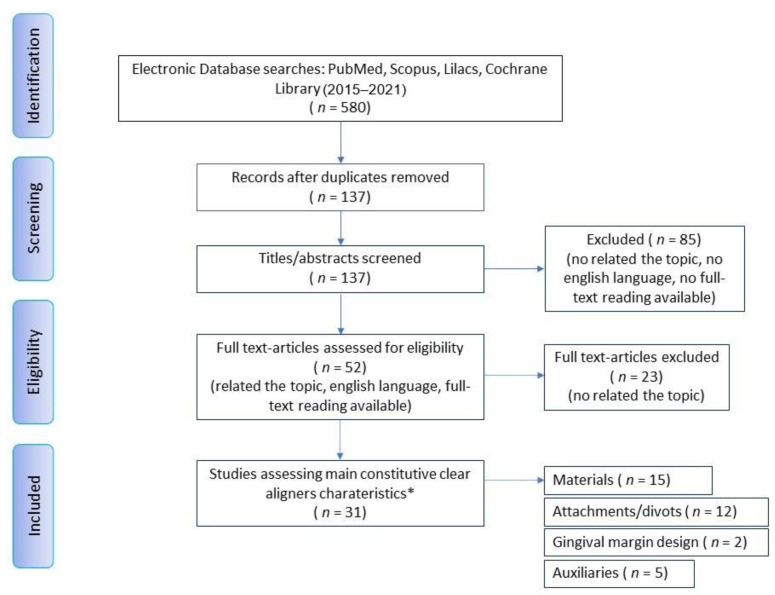
Review flow diagram. Caption: the PRISMA flow diagram for the systematic review detailing the database searches, the number of abstracts screened and the full texts retrieved. * Some articles analyzed > 1 characteristic.

**Table 1 ijerph-18-02870-t001:** Brands of clear aligners currently used in Italy with the main basic information and number of scientific studies in which they are used.

Name	Manufacturer	Material	Introduction	Daily/Weekly Wearing Time	Indications	Contraindications	Attachments	Divots	Gingival Margin Design	Auxiliaries	Studies
AirNivol	AirNivol	PU * or PET-G **	2010	22 h/14 days	Diastema, crowding, class II and III, deep bite, open bite, crossbite, preprosthetic and presurgical orthodontics	None	Yes	Yes	straight at the gingival zenith	Yes	3
ALL IN	Micerium S.p.A.	PET-G	2010	22 h/14 days	Diastema, crowding, open bite, cross bite, intrusion, extrusion, class I, II, III malocclusions	None	Yes	No	scalloped		1
ArcAngel	Network Gruppo Dextra	PET-G	2011	22 h/14 days	Diastema, crowding, rotation, intrusion/extrusion, orthodontic relapse	Class II and III need a surgical approach, severe cross bite, excessive buccal/lingual	Yes	No	scalloped	No	1
Clear Aligner	Scheu Dental	PET-G	2013	17 h/14 days	Diastema, crowding, rotation, torque control, orthodontic relapse	No patient compliance	Yes	Yes	straight 2 mm up the gingival zenith	Yes	10
Dair	Orthovit Devices	PU or PET-G	2010	23 h/14 days	Diastema, rotation, crowding	Severe malocclusion or periodontal problems	Yes	No	straight at the gingival zenith	No	0
Dental Stealth	Function Research srl	PU or PET-G	2008	22 h/15 days	Diastema, crowding, intrusion/extrusion, deep bite/open bite	None	Yes	No	scalloped	No	0
Effect Aligners	Orthofan	PET-G and PE ***	1998	22 h/14 days	Class I, II, III malocclusions, diastema, crowding, rotation, deep bite, open bite, asymmetry	No patient compliance or bad hygiene	Yes	No	scalloped	No	0
F22	Sweden & Martina Spa	PU	2010	22 h/14 days	All the malocclusions	No patient compliance	Yes	Yes	straight at the gingival zenith	Yes	9
Instaligner	CP Laboratorio Ortodontico	PET-G	2010	22 h/20 days	Diastema, crowding, intrusion/extrusion	TMJ **** disorders, severe skeletal problems	Yes	No	scalloped	No	0
Invisalign	Align Tech	Smart Track (multilayer aromatic thermoplastic polyurethane	1998	22 h/14 days	All the malocclusions	No patient compliance	Yes	No	scalloped	Yes	110
Inwisible	Wilocs	PET-G	2010	22 h/14 days	Diastema, crowding, rotation, intrusion/extrusion, light distal movements	Extractive cases, skeletal crossbites	Yes	No	scalloped	Yes	0
Nuvola	G.E.O. srl	PU	2006	20 h/10–15 days	Diastema, crowding, II Class malocclusion, cross bite, deep bite	Skeletal crossbites, severe vertical/trasversal problems	Yes	No	scalloped	Yes	2
Orthocaps	Ortho Caps Gmbh	Dual layer polymer 2006		20 h/14 days	All the malocclusions	No patient compliance	Yes	No	scalloped	Yes	0
Smart Evolution	Ortho Evolution srl	PE	2009	customized	All the malocclusions including those needing transversal palatal expansion	None	Yes	No	scalloped	No	0
Smile Clear	Orthodontics High Design	PET-G	2012	22 h/15 days	Crowding, crossbite, deep bite, open bite, orthodontic relapse, preprosthetic orthodontics	Class III malocclusions	Yes	Yes	scalloped	Yes	0
SmileLine	SmileLine	PET-G	2009	22 h/10–15 days	Light and moderate malocclusions	Severe malocclusions	Yes	No	scalloped	Yes	0
Smilers	Biotech Dental	PET-G	2014	22 h/15 days	Diastema, crowding, occlusal problems, TMJ disorders	Periodontal problems	Yes	No	scalloped	No	0
Smiletech	Ortodontica Italia srl	PU/PET-G	1999	22 h/15 days	Diastema, crowding, deep bite, open bite, preparation/finalization maxillo-facial surgery	None	Yes	No	scalloped	Yes	1
Sorridi	Tecnologia Dentale	PET-G	2015	22 h/7 days	Crowding, class I, II and III malocclusions, surgical-orthodontic cases	No patient compliance	No	Yes	straight 2 mm up the gingival zenith	Yes	1

Note: * PU = Polyurethane; ** PET-G = Polyethylene terephthalate glycol; *** PE = Polyester; **** TMJ = temporomandibular joint.

**Table 2 ijerph-18-02870-t002:** Articles included in the review.

Authors	Year	Country	Study Design	Study Size and Characteristics	Characteristic Investigated	Topic of the Study	Outcomes
Weir [12]	2017	Australia	Narrative Review	Not Applicable	attachments/gingival margin design/auxiliaries	Clear aligners in orthodontic treatment	Primary (constitutive) features of clear aligners should guide the clinician in the choice of the different clear aligner systems available
Hennessy et al. [14]	2016	Ireland	Narrative Review	Not Applicable	attachments	Clear aligners’ evolution	Clear aligners evolved in their primary features (first of all in attachment design and indications) but they cannot ever be used interchangeably to fixed labial appliances
Lombardo et al. [15]	2018	Italy	Case report	1 hyperdivergent male patient (18 years old) with a Class II malocclusion from mandibular retrusion	auxiliaries	Class II subdivision correction with clear aligners using intermaxillary elastics	Combining F22 aligners with appropriate auxiliaries is an efficacious means of resolving orthodontic issues such as class II, dental crossbite, and crowding in a time-frame comparable to that of conventional fixed orthodontics
Liu et al. [16]	2018	China	Comparative/in vitro study	5 sets of clear aligners (G0 aligners as a control group, with no activation; G1 aligners for intruding canines; G2 aligners for intruding incisors; G3 aligner for intruding canines and incisors with the same activations; G4 aligners for intruding canines and incisors with different activation)	attachments	Force changes associated with different intrusion strategies for deep-bite correction by clear aligners	With the same activation (0.2-mm intrusion) and rectangular attachments placed on premolars and first molars, canines experienced the largest intrusive force when intruded alone using G1 aligners. The canines received a larger intrusive force than incisors in G3. The incisors received similar forces in G2 and G3. First premolars endured the largest extrusive forces when all anterior teeth were intruded with G3 aligners. Extrusion forces were exerted on canines and lateral incisors when using G4 aligners
Caruso et al. [17]	2019	Italy	Retrospective study	Lateral cephalometric radiographs of 10 subjects (8 females and 2 males; mean age 22.7 ± 5.3 years) with class II malocclusion treated with Invisalign and no extractions	attachments	Impact of molar teeth distalization with clear aligners on occlusal vertical dimension	Upper molar distalization with orthodontic aligners guarantees an excellent control of the vertical dimension representing an ideal solution for the treatment of hyperdivergent or openbite subjects
Lombardo et al. [27]	2020	Italy	Comparative/in vitro study	1 dental cast of a patient (class I malocclusion, no caries, no recessions, no prosthesis)	material/gingival margin design	Micro Computer Tomography X-ray comparison of aligner gap and thickness of six brands of aligners (Invisalign, Nuvola, F22, AirNivol, Arc Angel, ALL IN)	There are differences between the six aligner systems examined in terms of 2D and 3D measurements of aligner thickness and gap
Bucci et al. [28]	2019	Italy	Prospective clinical study	13 F, 5 M (28.8 ± 9.6 years)	material	Thickness of orthodontic clear aligners (AirNivol) after thermoforming and after 10 days of intraoral exposure: A prospective clinical study	Passive and active (with attachments and divots) clear aligners examined have good thickness stability after intraoral ageing
Jindal et al. [29]	2020	India/UK	Comparative/in vitro study	3 dental casts of a class I malocclusion	material	Mechanical behavior of 3D printed vs. thermoformed clear dental aligner materials under non-linear compressive loading using Field Emission Microscopy	Material and technique of clear aligner production show comparable mechanical behavior
Porojan et al. [30]	2020	Romania	Comparative/in vitro study	42 thermoformed samples from 3 thermoplastic materials for clear aligners (Biolon, Crystal, Duran)	material	Surface quality of thermoplastic materials for clear aligners after beverages and cleaning agents exposure	Biolon material has demonstrated the most constant behavior compared to Crystal and Duran
Ihssen et al. [31]	2019	Germany	Comparative/in vitro study	60 specimens of CA Clear Aligner (immersed in distilled water; subjected to accelerated ageing and used like control)	material	Effect of in vitro aging by water immersion and thermocycling on the mechanical properties of PETG aligner material	Intraoral temperature alternating to water absorption promotes a degradation of orthodontic aligners with a decrease in orthodontic forces
Lombardo et al. [32]	2017	Italy	Comparative/in vitro study	4 specimen sheets (F22 clear aligner, Duran, Erkoloc Pro, Durasoft)	material	Stress relaxation properties of four orthodontic aligner materials	Duran and F22 are more stiff than the double layer materials. F22 yielded the greatest initial stress values but also high velocity of decay. Duran presented the higher velocity of stress relaxation. Durasoft had the smallest decay.
Alexandropoulos et al. [33]	2015	Greece	Comparative/in vitro study	8 clear aligners (four thermoplastic materials: Clear Aligner, ACE, A+, Align Technology)	material	Chemical and mechanical characteristics of contemporary thermoplastic orthodontic materials	Invisalign (Align Technology) showed higher hardness and modulus values, a slightly higher brittleness and lesser creep resistance compared with the PETG-based products
Elkholy et al. [34]	2017	Germany	Comparative/in vitro study	3 mandibular clear aligners made with Duran thermoplastic material with different thicknesses (0.5, 0.625, 0.75 mm)	material	Mechanical load exerted by PET-G aligners during mesial and distal derotation of a mandibular canine	The 0.625 and 0.75 mm aligners have similar mechanical behavior. Derotation of lower canines should be limited to 10°
Putrino et al. [35]	2020	Italy	Prospective study	100 patients (57 F, 43 M, age 7–46) with fixed appliances, removable appliances, clear aligners (Invisalign, F22, Smiletech and Sorridi)	divots	The management of orthodontic therapies during the pandemic	Clear aligner treatments are the most comfortable and efficient. Clear aligners without attachments and equipped with divots and straight margin showed the best behavior
Lombardo et al. [36]	2018	Italy	Case report	Female patient (23 years old) with a Class III malocclusion, transverse maxillary deficiency and bilateral crossbite	auxiliaries	Class III malocclusion and bilateral cross-bite in an adult patient treated with miniscrew-assisted rapid palatal expander and aligners	The combined therapy with a novel miniscrew-assisted rapid palatal expander and F22 aligners allowed the successful treatment of the case
Lombardo et al. [37]	2020	Italy	Case report	Male patient (16 years old) with Class II malocclusion and maxillary transverse skeletal deficiency	auxiliaries	Class II subdivision with skeletal transverse maxillary deficit treated by single-sitting bone-borne appliance	The combined therapy of a bone-borne palatal expander with miniscrews and the F22 aligners allowed the successful treatment of the patient in an acceptable timeframe
Tepedino et al. [38]	2018	Italy	Retrospective study	Digital models (pre-treatment, post-treatment and the digital setup) of 39 adult patients	attachments	Movement of anterior teeth (torque values) using clear aligners	No statistically significant difference was found for all the anterior teeth between predicted and achieved torque movements
Gerard Bradley et al. [39]	2016	Switzerland	Comparative/in vitro study	50 specimens obtained from 25 clear aligners (Invisalign) used for 44 ± 15 days from a patient and 25 never-used clear aligners utilized as reference	material	The mechanical and chemical properties of Invisalign appliances after use	Intraoral aging affects mechanical properties of the Invisalign appliance despite the lack of detectable chemical changes
Tamburrino et al. [40]	2020	Italy	Comparative/in vitro study	Circular foils of 3 materials (Duran, Biolon, Zendura) without thermoforming, after thermoforming and after thermoforming plus storage in artificial saliva	material	Mechanical properties of thermoplastic polymers for aligner manufacturing	Elastic modulus ever increases after thermoforming except for Biolon. The tensile yield stress, also after storing in artificial saliva, increases after thermoforming for Duran and decreases for Biolon and Zendura
Liu et al. [41]	2016	China	Comparative/in vitro study	3 clear aligners from a subject manufactured by 3 companies (Invisalign-Align Technology; Angelalign-EA Medical Instruments; Smartee-Smartee Denti-Technology)	material	Color stabilities of three types of orthodontic clear aligners exposed to staining agents	The Invisalign aligners were more prone to pigmentation than the Angelalign and Smartee aligners
Bernard et al. [42]	2020	Canada	Comparative/in vitro study	300 specimens (100 per brand) from clear aligners of 3 different brands (Invisalign, Clear Correct, Minor Tooth Movement)	material	Colorimetric and spectrophotometric measurements of orthodontic thermoplastic aligners exposed to various staining sources and cleaning methods	The Invisalign aligners were more prone to pigmentation than the ClearCorrect or the Minor Tooth Movement devices after exposure to coffee or red wine. Black tea caused important stains on the surface of the three tested brands.
Papadopoulou et al. [43]	2019	Switzerland	Comparative/ in vitro study	40 Invisalign aligners with attachments used (20 for 1 week, 20 for 2 weeks) from different patients and 10 Invisalign unused aligners	material	Changes in Roughness and Mechanical Properties of Invisalign^®^ Appliances after One- and Two-Weeks Use	Ageing has a detrimental effect on the surface roughness and mechanical properties of Invisalign appliances after 1 week of clinical usage
Edelmann et al. [44]	2020	USA	Comparative/digital	60 clear aligners 3D printed with 2 different resins (Dental LT and Grey V4) in three thicknesses (0.5, 0.75, 1 mm- 10 for each thickness value)	material	Analysis of the thickness of 3-dimensional-printed orthodontic aligners	3D-printed aligners were thicker overall than the corresponding design file. The Dental LT aligners had the largest thickness deviation, whereas the Grey V4 without spray had the smallest. Increased thickness may deleteriously affect the clinical utility of clear aligners
Iliadi et al. [45]	2019	Greece/Switzerland	Systematic Review and Meta-Analysis	13 studies in vitro describing aligner thickness	material	Forces and moments generated by aligner-type appliances for orthodontic tooth movement: A systematic review and meta-analysis	Aligner thickness does not appear to possess a significant role in forces and moments generated by clear aligners under specific settings, while the most commonly examined tooth movements are tipping and rotation.
Barreda et al. [46]	2017	Argentina	Prospective study	10 subjects (15–50 years old) with mild or moderate upper crowding, on whose teeth 40 attachments were applied	attachments	Surface wear of resin composites used for Invisalign attachments	The alteration of the attachment surface during the first six months of treatment depends on the composite used, while attachment shape does not appear to be affected
Costa et al. [47]	2020	Brazil	Comparative/digital	3 clear aligners obtained from 3 prototypes of maxillary models (each one with a specific attachment with different geometry on the central incisor to guide extrusion)	attachments	Effect of three different attachment designs on the extrusive forces generated by thermoplastic aligners in the maxillary central incisor	The attachment geometry designed with a frontal face without edges and less protrusive, with a vestibular length of 3.32 mm, showed best distribution of forces for extrusion movement compared to the others
Dasy et al. [48]	2015	USA	Comparative/ in vitro study	12 types of aligners with different thicknesses (soft, medium, hard, and Essix ACE^®^ for retainer) were obtained from 3 casts (two with ellipsoid and beveled attachments and one without any attachment as a control)	attachments	Effects of variable attachment shapes and aligner material on aligner retention	Ellipsoid attachments had no significant influence on the force required for aligner removal and hence on aligner retention. Essix ACE^®^ showed significantly less retention than CA^®^-hard on the models with attachments. Beveled attachments were observed to increase retention significantly, compared with ellipsoid attachments and when using no attachments
Staderini et al. [49]	2020	Italy	Case series	2 patients (8 years old) with anterior crossbite (−1 mm negative overjet), Class I (into a tendency towards Class III)	attachments	Indication of clear aligners in the early treatment of anterior crossbite	Overjet and overbite were corrected in both patients in 5 months of treatment with clear aligners; bite ramp attachments are useful to correct anterior crossbite
Weckmann et al. [50]	2020	Germany	Comparative/in vitro study	2 attachments (ellipsoid and rectangular) bonded 30 times on a master dental cast with different protocols and composites	attachments	Influence of attachment bonding protocol on precision of the attachment in aligner treatments	The bonding protocol with high viscous composite without a perforation in the attachment reservoir is inaccurate. The use of a low viscous composite or attachments made by a two-phase procedure with high viscous composite revealed more precise results
Mencattelli et al. [51]	2015	Italy	Comparative/in vitro study	2 types of invisible aligners to analyze, respectively, a malocclusion with a high maxillary canine, and the effects on the axial rotation of a maxillary central incisor with and without a divot	divots	Novel universal system for 3-dimensional orthodontic force-moment measurements and its clinical use	The efficacy of using invisible aligners with a divot was validated
Patterson et al. [52]	2021	USA	Prospective study	80 adult patients (Group 1 with Class I molar malocclusions; (11 men and 29 women); 38.70 ± 15.90 years) and Group 2 with Class II molar malocclusions (11 men and 29 women; 35.25 ± 15.21 years)) under Invisalign treatment	auxiliaries	Class II malocclusion correction with Invisalign: is it possible?	The Invisalign system successfully achieves certain tooth movements but fails to achieve others predictably. No significant Class II correction or overjet reduction was observed with elastics for an average of 7-month duration in the adult population. Additional refinements may be necessary to address problems created during treatment (posterior open bite)

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
