# Peer review of "Clear Aligners: Between Evolution and Efficiency—A Scoping Review"

_ijerph, 2021, doi:10.3390/ijerph18062870_

Round 1

Reviewer 1 Report

This manuscript is well written. However, to match the topic, some specific comments were listed below

1. Materials and methods/Results: the authors mentioned the main constitutive characteristics of clear aligners systems are the materials used, the design of the gingival margin, the presence of attachments. Please provide the materials of Invisalign in Table 1.

2. Please try to clarify the treatment "efficiency" of clear aligner in different materials, gingival margin design, and different attachment design in discussion.

3. In table 1, some brands mentioned " no patient's compliance" was contraindication. However, it seems " no patients' compliance" was applied to all brands.

This paper titled “Clear Aligners: Between Evolution and Efficiency.” Review of the literature. 31 studies published between 2015 and 2021 were included in the scoping review. This review is relevant valuable, but there were some errors that need editing.

Specific:

a. Figure 1, Electronic Database searches:…….(2015-2020)-- (2015-2021)

b. Fig 2 is vague, please change it.

c. Reference: please follow the style of reference in instruction to authors: 1. Author 1, A.B.; Author 2, C.D. Title of the article. Abbreviated Journal Name Year, Volume, page range. Especially ref. 1, 2, 8.

Reviewer 2 Report

To review the evolution of clear aligners based on their characteristics with focus placed specifically on the verification of scientific evidence currently available on existing clear aligners based on the material they are made of, the gingival margin design, the presence of attachments with 45 different shapes, the alternative use of divots and the use of auxiliaries tools (i.e. elastics, 46 mini-screws, etc.).

The topic is interesting and relevant to the modern dentistry and orthodontic research. The review of literature is up to date and reflects research and finding published in many academic journals within the last 5 years. The topic is original and suitable for publication due to the fact that this systematic literature review summarises the areas of development and progress made by researchers in this sector point out the key findings and noteworthy pieces of work to consider.

This paper provides insight and combine the most recent contributions scientific literature on the basic constitutive characteristics of invisible aligners; thus demonstrating that the knowledge of experts converges in classifying the behaviour of materials and their mechanical properties allowing recommendations to be made based on the type of equipment to be used. Key recommendations made also contribute to current/new knowledge, but this paper provides a rationale argument based on a systematic review process.

The use of English language throughout the paper is good, however there are a few minor grammatical, spelling, tense and English language errors that by proof reading again would no doubt be resolved (this mainly occurs at the start of sentences or paragraphs). The overall writing of the paper is clear and easy to read even with some errors, therefore through additional proof reading and corrections, this will allow the reader to easily understand the content presented.

Although the conclusions are short, they accurately reflect the research presented and findings discussion in the results and conclusions.

Conclusions address the main question posed, however the authors do recognise the limitations of their work highlight that one aspect of the research question on “gingival margin” is not an element where extensive research is available right now. Critically and accurately due to limited research in some of the areas discussed, the author highlight that the results present few comparative studies between systems of invisible aligners that place the attention on the need to deepen this are of knowledge through experimental studies to aid informed decisions that can be made in practice.

The quality of the images provided and observed during this review are of a low-res standard. Please ensure that images of appropriate resolution are provided as this does make the diagrams some what difficult to read.

The abstract although well written does omit one key piece of information which is the overview of the brands of aligners used in Italy and its relevance to the paper.

The methodology used is appropriate and the search terms logical. It would be interesting to know whether the systematic review has been registered with Prospero or not. Although the paper is not directly related to human or animal systematic reviews, the equipment being assessed is. If the authors feel this is appropriate for a Prospero registration, they should be encouraged to do this.

Please provide the date/date period when the systematic review was conducted to allow the readers to establish why articles that may have been published in the year period (2015-2020) defined may not have been reviewed (i.e. may have been in-press and not available at the time when the search was conducted but still published in the time period). 

Although a detailed criteria has been established, there are a few articles from existing literature that maybe need to be included in the results. I would therefore encourage the authors to double check whether any of the following articles should be included in the relevant within the list of articles reviewed or as referenceable literature used to provide context to the papers topic/introductory context:

  • DietrichC.A.EnderA.BaumgartnerS. and MehlA. (2017), “A validation study of reconstructed rapid prototyping models produced by two technologies”, The Angle Orthodontist, Vol. 87 No. 5, pp. 782-787.
  • Jindal, P., Juneja, M., Bajaj, D., Siena, F.L. and Breedon, P., 2020. Effects of post-curing conditions on mechanical properties of 3D printed clear dental aligners. Rapid Prototyping Journal.
  • Liu, Y.F., Zhang, P.Y., Zhang, Q.F., Zhang, J.X., Chen, J., 2014 Jan 1. Digital design and fabrication of simulation model for measuring orthodontic force. Bio Med. Mater. Eng. 24 (6), 2265–2271.
  • MartorelliM.GerbinoS.GiudiceM. and AusielloP. (2013), “A comparison between customized clear and removable orthodontic appliances manufactured using RP and CNC techniques”, Dental Materials, Vol. 29 No. 2, pp. e1-e10.
  • McCarty, M.C., Chen, S.J., English, J.D. and Kasper, F., 2020. Effect of print orientation and duration of ultraviolet curing on the dimensional accuracy of a 3-dimensionally printed orthodontic clear aligner design. American Journal of Orthodontics and Dentofacial Orthopedics158(6), pp.889-897.
  • Naeem, O.A., 2020. A Comparison of Three-Dimensional Printing Technologies on the Precision, Trueness, and Accuracy of Printed Retainers.

Although some of the recommended papers above may fall outside of the search parameters (2015 - 2020), please do not overlook the need to provide some context on the developments made leading up to the search period, otherwise the reader will not know whether the developments made are significant or insignificant for the time period reviewed. The major developments leading up to 2015 would benefit from more detail information providing greater context to the first time reader i.e. what are the major developments over the past 20 years and what impact have they had in the lead up to the 2015-2020 time period reviewed.

The use of English language throughout the paper is good, however there are a few minor grammatical, spelling, tense and English language errors that by proof reading again would no doubt be resolved (this mainly occurs at the start of sentences or paragraphs) for example "A preliminary research of the clear dental aligners systems...." should read "Preliminary research on clear dental aligner systems....".

Reviewer 3 Report

I am not clear on the aim or on the type of review this is intended to be. It does not conform to the standards of either a general literature review or of a scoping review. The reporting of the study is not appropriate for either type. It is too general and non-specific for a scoping review and does not conform to the PRISMA standards of reporting for a scoping review (see Equator Network for further information). It is does not give a broad enough general overview as would be needed for a general literature review. The introduction and rationale do not provide an understanding of the need to do this work. The aim is not clear at the end of the introduction. The reporting of the methods is not clear. The reporting of the results is also poor, for example a characteristics table is included that is little more than a list of references with a very basic description of the focus of the study and none of the information that is normally expected (study design, study size, location, participant characteristics or appropriate equivalent etc...)

Some specific recommendations: 

Page 1 line 28 – confusing to list Europe and Italy as if Italy were outside of Europe. Suggest either just ‘Europe’ or ‘Italy and the rest of Europe’ or ‘Italy and other European countries’ or similar.

Page one sentence starting on line 35 continues to line 39 – this is a very long sentence which makes it difficult to follow – suggest minor edits to improve readability/clarity

The aim of the review is confusing – you either want to give a historical overview and context or you want to establish the effectiveness of the intervention. It is not possible to do both here. Given the nature of this review not being a systematic review, I would suggest the first aim is most appropriate. You may be able to discuss the implications for effectiveness in the discussion section based on your review findings but you should not mention effectiveness in your aim.

First sentence of methods section is 7 lines long – this is very difficult to read and follow. Please shorten sentence.

In the methods section the review is described as a scoping review rather than just a literature review – you should be consistent in description throughout.

The research question is actually 2 questions, it might be simpler to separate these. Also this should be described at the end of the introduction section with the aim.

Eligibility criteria – it sounds like no limit was put on study design? May be simpler to state this.

Why was a limitation put on year of publication?

You should report on the exact structure and combinations of the search strategy for a scoping review – so that it is reproducible. Giving a list of key words is not sufficient (though it would be if this was just a literature review).

Starting line 77 - I am unclear on what is meant by ‘the authors analysed the information’ – are you here referring to data extraction and that this was done in duplicate by 2 independent reviewers? If so, please state this for clarity.

Line 83 – not clear on the review being structured according to PRISMA flow diagram – perhaps what is meant is that the results of the search are depicted in the PRISMA flow diagram? (please note that PRISMA is an abbreviation and should be capitalised)

Results section – not clear why you are describing the content of papers that you screened out. If you have 137 papers after duplicates are removed, how can you have 390 papers about Invisalign? Are you counting duplicates?  

Line 97 is very unclear. I do not understand what is being communicated.

PRISMA flow diagram and Figure 2 are poor resolution and needs to be improved
